METHODS AND RESOURCES

# SPOT: A machine learning model that predicts specific substrates for transport proteins

**Alexander Kroll**[1], **Nico Niebuhr**[1], **Gregory Butler**[2], **Martin J. Lercher** [1]*

**1** Institute for Computer Science and Department of Biology, Heinrich Heine University, Düsseldorf, Germany, **2** Department of Computer Science and Software Engineering, Concordia University, Montreal, Quebec, Canada

* martin.lercher@hhu.de

**Data Availability Statement:** The code used to generate the results of this paper, in the form of Jupyter notebooks, is available at https://github.com/AlexanderKroll/SPOT and at https://doi.org/10.5281/zenodo.13234753. All datasets used to

## Abstract

Transport proteins play a crucial role in cellular metabolism and are central to many aspects of molecular biology and medicine. Determining the function of transport proteins experimentally is challenging, as they become unstable when isolated from cell membranes. Machine learning-based predictions could provide an efficient alternative. However, existing methods are limited to predicting a small number of specific substrates or broad transporter classes. These limitations stem partly from using small data sets for model training and a choice of input features that lack sufficient information about the prediction problem. Here, we present SPOT, the first general machine learning model that can successfully predict specific substrates for arbitrary transport proteins, achieving an accuracy above 92% on independent and diverse test data covering widely different transporters and a broad range of metabolites. SPOT uses Transformer Networks to represent transporters and substrates numerically. To overcome the problem of missing negative data for training, it augments a large data set of known transporter-substrate pairs with carefully sampled random molecules as non-substrates. SPOT not only predicts specific transporter-substrate pairs, but also outperforms previously published models designed to predict broad substrate classes for individual transport proteins. We provide a web server and Python function that allows users to explore the substrate scope of arbitrary transporters.

## Introduction

Transport proteins are fundamental components of cellular processes. They shuttle small molecules, ions, and macromolecules across biological membranes, facilitating the flow of specific molecules across cells and cellular compartments. Despite the importance of these proteins, a large fraction of transporter genes lack high-quality functional annotations even for well-studied model organisms [1,2]. Determining the functions of insufficiently characterized transporters is critical for the construction of more precise and comprehensive models of cellular metabolism. Knowledge of the substrate specificities and mechanisms of action of transporters can also provide valuable information for designing and engineering cellular systems, e.g., for the production of biofuels and pharmaceuticals [3,4]. In addition, transport proteins can be

generate the results of this manuscript are available at https://doi.org/10.5281/zenodo.8358682.

**Funding:** This work was funded through grants to MJL by the European Union (ERC AdG "MechSys"–Project ID 101055141) and by the Deutsche Forschungsgemeinschaft (DFG, German Research Foundation: CRC 1310, and, under Germany's Excellence Strategy, EXC 2048/1–Project ID675 390686111). The funders did not play any role in the study design, data collection and analysis, decision to publish, or preparation of the manuscript.

**Competing interests:** The authors have declared that no competing interests exist.

**Abbreviations:** CV, cross-validation; ECFP, extended-connectivity fingerprint; FCNN, fully connected neural network; GO, Gene Ontology; HMM, hidden Markov model; MCC, Matthew's correlation coefficient; NLP, natural language processing; PPV, positive predictive value; SVM, support vector machine; TCDB, Transporter Classification Database.

targeted for drug delivery [5], providing opportunities for the development of new therapeutics.

Determining the function of transport proteins experimentally is challenging. They are characterized by a partially hydrophobic surface and are hence unstable when isolated from cell membranes, making their purification and in vitro study difficult. Consequently, experimental 3D structures for transmembrane proteins are approximately 4 to 5 times underrepresented compared to non-transmembrane proteins [6]. Although recent advances with models such as AlphaFold [7], ESMFold [8], and RoseTTAFold [9] allow accurate prediction of the 3D structure of most transport proteins, there is no method that can use the predicted protein structure to infer protein function with higher accuracy than sequence-based models. As a result, high-quality functional annotations are available for less than 1% of sequenced transport proteins [2].

The difficulty of experimental characterizations makes the development of suitable computational methods highly desirable. The most widely used computational approaches for inferring the functions of transport proteins search for related transport proteins in protein databases either via sequence similarity or via profile hidden Markov models (HMMs) [10–14]. However, such methods can only deliver satisfactory results if highly similar homologous transporters with known functions exist.

In comparison to similarity-based approaches, recent machine learning-based methods have shown superior performance in the prediction of the class of molecules transported by a given transporter [15–19]. These methods utilize features extracted from the protein amino acid sequence as the input for neural networks, RBF networks, or support vector machines (SVMs). Because SVMs are based on the similarity or distance between pairs of data points, they are essentially similarity-based models as well. Currently, the state-of-the-art tools for predicting substrate classes are 2 SVM-based models, TooT-SC [19] and TranCEP [18]. TooT-SC was trained on 1,524 data points to predict one out of 11 substrate classes for every transporter, achieving an accuracy of 82.5% on independent test data. TranCEP, on the other hand, was trained on 900 data points and predicts one out of 7 substrate classes (cations, anions, sugars, amino acids/oligopeptides, proteins/mRNA, electrons, and other substrates), achieving an accuracy of 74.2%. The accuracies of both models cannot be directly compared due to differences in the training tasks and data sets.

A promising approach to the prediction of transport protein functions could be the application of Transformer Networks, deep neural networks originally developed for modeling sequences [20]. While Transformer Networks were originally developed for natural language processing (NLP) tasks, they can also be trained to process sequences from other modalities, such as protein amino acid sequences [21] or molecular structures [22]. However, the only previous application of Transformer Networks to the functional classification of transporters is a model of much more limited scope than those discussed above, providing predictions for only 12 specific inorganic anion or cation substrates [23].

All previous models are limited in their utility, as they require the existence of similar transporters with known function, they are designed to predict broad classes of substrates rather than specific substrates, or they were trained specifically for small groups of substrates and the corresponding transporter families. Additionally, previous methods often utilized suboptimal approaches for generating numerical representations of protein sequences, and they were trained on relatively small data sets of only 900 to 4,100 data points, despite the availability of much more data in protein databases. As yet, no comprehensive general model capable of accurately predicting specific substrates across various transporter families exists.

In this study, our goal was to develop such a prediction model. We aimed to predict whether a given molecule is a substrate for a particular transport protein from the molecular

structure of the molecule and the linear amino acid sequence of the protein, without relying on the protein's 3D structure. Although 3D structures can be predicted for most proteins [7–9], current deep learning tools can extract functional information much more easily from the amino acid sequence than from the 3D structure [24]. We generated highly informative transporter and substrate representations by using 2 Transformer Networks trained to process protein amino acid sequences [21] and string representations of small molecules [22], respectively. To train these models for the prediction of specific substrates and substrate classes for transport proteins, we generated a data set substantially exceeding previously used data sets for related tasks.

## Results

### Construction of an experimentally validated transporter-substrate data set

To construct the data set, we extracted 4,775 transporter-substrate pairs with experimental evidence from the Gene Ontology (GO) database [25] and 15,864 data points from the UniProt database that were labeled as manually reviewed [2]. After removing 2,633 duplicates of data points present in both databases, the resulting merged data set consisted of 18,006 unique transporter-substrate pairs with UniProt IDs for the transporters and ChEBI IDs for the substrates.

The Transporter Classification Database (TCDB) [26] is another database containing functional information for transport proteins. Examining a random sample of literature sources listed in TCDB entries, we noticed that many assigned protein functions have not been validated experimentally. To further investigate the quality of the TCDB data, we extracted 18,051 transporter-substrate pairs with UniProt IDs from the TCDB, covering 11,799 different proteins. Proteins in the UniProt database have been assigned annotation scores that range from 1 to 5 and provide a measure of accuracy of the protein annotation; the extraction of transporter-substrate pairs from UniProt described above considered only those with score 5. In constrast, over 50% of UniProt IDs extracted from the TCDB have a UniProt annotation score of 2 or below (S1 Fig), indicating a lack of experimental evidence for many TCDB entries. To ensure high data quality, we thus did not use data from TCDB.

For the binary task of predicting whether a given molecule is a substrate for a transporter, we mapped all substrates in the data set to InChI (International Chemical Identifier) [27] and SMILES [28] strings, which are textual identifiers of the structure of chemical substances. Some of the substrates could not be mapped, e.g., because their ChEBI IDs were referring to general molecule typess—such as sugar, ion, or protein—instead of specific molecules. We removed the corresponding data points, 78% of which were extracted from the UniProt database and 22% from the GO Annotation database. Furthermore, we removed all ~7,500 transporter-substrate pairs with a proton as a substrate. We hypothesized that for the majority of these data points, the transport protein is either a symporter or an antiporter with the proton not being the primary substrate. This resulted in a final data set of 8,587 data points, comprising a diverse set of 5,882 distinct transport proteins and 364 unique substrates. The set of transport proteins includes channel transporters, electrochemical potential-driven transporters, primary active transporters, and group translocators, and the 364 substrates present in our data set are distributed across 7 diverse substrate classes (S1 Data lists all substrates, their number of occurrences, and their substrate class).

We split the resulting data set into 80% training data and 20% test data, ensuring that no protein in the test set also occurs in the training set. Such random splits can result in misleading performance metrics because many highly similar data points could be included in both the training and test sets. For this reason, we evaluated model performance not only on the

entire test set, but also for different levels of sequence identity between test and training proteins and for different substrate occurrences in the training set. This strategy provides systematic estimates of how prediction quality depends on the similarity of a data point to the training data points.

To generate a second data set for predicting the substrate classes of transport proteins, we followed earlier publications by classifying each substrate in our data set into one of 7 categories: cations, anions, sugars, amino acids/oligopeptides, proteins/mRNA, electrons, and other substrates [16–18]. In cases where we lacked sufficient information to classify a substrate or where we encountered multiple substrate classes for a given transport protein, we discarded those data points for the classification task (for more details, see Methods). Ultimately, this resulted in a data set with 11,664 transporter amino acid sequences, each mapped to one of the 7 substrate classes. As for the data set on transporter-substrate pairs, we divided this data set into 80% training data and 20% test data in such a way that no protein from the test set would also occur in the training set, and we calculated the maximal pairwise sequence identity for each test protein compared to all proteins in the training set [29].

## Sampling negative data points

Training a prediction model capable of distinguishing between substrates and non-substrates for a given transporter also requires negative data, i.e., transporter-molecule pairs where the molecule is not a substrate of the transport protein. However, such negative data is not systematically recorded in protein databases. We hypothesized that negative data points could be artificially generated through random sampling, a common strategy in classification tasks without negative training data [30]. To ensure that the model learns to distinguish substrates from structurally similar non-substrates, we sampled negative training data preferably from molecules structurally similar to a known true substrate of the transporter, a strategy that proved successful in the prediction of enzyme-substrate pairs [31]. However, we only considered those 364 molecules included among the experimentally confirmed transporter-substrate pairs in our data set. We assumed that this subset of substrates mostly comprises molecules that are likely to occur in biological cells. Although transport proteins can be promiscuous [32], among such a limited and specific subset, most of the potential secondary substrates are likely not included for the majority of transporters.

Moreover, it is more likely for 2 similar transport proteins to share the same set of substrates. While proteins with amino acid sequence identities between 40% and 60% are still likely to be structurally similar [33], functional similarity is much less likely below a threshold of approximately 60% [34]. When randomly sampling negative (non-substrate) small molecules for a given transporter, there is a small chance of sampling false negative data points, i.e., assigning transporter-small molecule pairs to the negative class that are in fact true positive transporter-substrate pairs. To reduce the chance of accidentally sampling such false negative data points, when sampling non-substrates for a given transporter, we excluded all molecules that occurred as a true substrate for any transport protein in our data set that are highly similar to the given transporter (i.e., that have an amino acid sequence identity >60%; for more details, see Methods). This choice was made to ensure that our training data set does not contain too many false negative data points, which would likely result in reduced model performance, and was not motivated by a desire to make the prediction task easier. In contrast, as described above, we attempted to make the prediction task more difficult by sampling negative metabolites that are structurally similar to the true positive substrates. Setting this threshold to a much lower value than 60% would have led to the exclusion of substrates from distantly related proteins with likely distinct substrates, while selecting a higher threshold might lead to

sampling negative molecules from the substrates of highly similar proteins, potentially resulting in a higher number of false-negative data points. Given this carefully chosen sampling process, we assumed that the frequency of incorrectly generated negative labels is sufficiently low to not adversely affect model performance. This assumption was confirmed a posteriori by the high model accuracy on independent test data, as detailed below.

We chose to generate more negative data points than we have positive data points, as this more closely reflects the true distribution of positive and negative transporter-molecule pairs compared to a balanced distribution, and as it has the added benefit of providing more data for training and testing. Specifically, for each positive transporter-substrate pair, we randomly sampled between 1 and 4 negative data points. On average, we obtained 3 negative data points for every positive transporter-substrate pair. We found that sampling a variable number of negative data points led to better generalization capabilities of the resulting model compared to a fixed ratio of positive to negative data points for each transporter. The sampling process resulted in a final data set consisting of 33,162 data points.

## Representing substrates as numerical vectors

Machine learning algorithms require input in the form of numerical vectors. Thus, we need to numerically represent information about the transporters and the candidate substrates. Most substrates are small molecules, which are typically represented numerically either through expert-crafted fingerprints or through deep learning-based representations. We chose to calculate both types of representations, learned and expert-crafted, for all substrates in our data set to compare their performance.

Expert-crafted fingerprints are bit vectors that are generated by applying expert-chosen functions to calculate a representation of the small molecule. The most commonly used such fingerprint is the extended-connectivity fingerprint (ECFP), also known as Morgan fingerprint [35]. To calculate ECFPs, molecules are represented as graphs by interpreting the atoms as nodes and the chemical bonds as edges. Bond types and feature vectors with information about every atom are calculated (types, masses, valences, atomic numbers, atom charges, and number of attached hydrogen atoms) [35]. Afterwards, these identifiers are updated for a predefined number of steps by iteratively applying expert-chosen functions that summarize aspects of neighboring atoms and bonds. After the iteration process, all identifiers are used as the input of a hash function to produce a binary vector with structural information about the molecule. The number of iterations and the dimension of the fingerprint can be chosen freely. We set them to the default values of 3 and 1,024, respectively; lower or higher dimensions led to inferior predictions. We calculated 1,024-dimensional ECFPs for all substrates in our data set using their InChI strings as the input.

To generate learned substrate representations, we applied the ChemBERTa Transformer Network [22]. ChemBERTa processes SMILES strings [28], which are string representations that summarize the molecular structure of small molecules. Before being processed, every SMILES string is divided into tokens, which are small chunks of the input sequence. The ChemBERTa model was trained on 77 million unique SMILES strings for the self-supervised task of predicting the identity of randomly masked tokens in the input sequence. This training procedure forces the model to store information about the molecule in a 767-dimensional representation vector for each individual input token. After model training, a single fixed-length numerical representation of the molecule can be extracted by calculating the element-wise mean across all token representations. We calculated these numerical molecule representations, in the following referred to as ChemBERTa vectors, for all substrates in our data set.

## Representing transport proteins as numerical vectors

We calculate transport protein representations through the ESM-1b model [21], a state-of-the-art Transformer Network trained on ~27 million proteins from the UniRef50 data set [36]. ESM-1b takes an amino acid sequence as its input and outputs a 1,280-dimensional numerical representation of the sequence. Every amino acid in the protein sequence is represented through a separate token. During model training, a randomly chosen ~15% of the amino acids of every input sequence are masked, and the model is trained to predict the identity of the masked amino acids. This training procedure forces the model to store both local and global information about the protein sequence in one 1,280-dimensional representation vector for each individual amino acid. In order to generate a single fixed-length numerical representation of the whole protein, one typically calculates the element-wise mean across all amino acid representations [21,37,38]. We refer to these compressed protein representations as ESM-1b vectors.

In contrast to these general protein representations, which are not specifically adjusted for the task of interest, we additionally generated task-specific ESM-1b vectors by fine-tuning the ESM-1b model for the transporter-substrate prediction task (**S1 Text**). In the following, these representations will be referred to as ESM-1b$_{ts}$ vectors.

## Using combined transporter and molecule representations leads to successful predictions of transporter-substrate pairs

For a given transporter-molecule pair, our aim was to predict if the molecule is a substrate for the transporter. To solve this binary classification task, we concatenated a transporter representation (ESM-1b or ESM-1b$_{ts}$ vector) with a molecule representation (ECFP or ChemBERTa vector) for all transporter-molecule pairs in our data set (**Fig 1**). To compare the performances of the different representations, we used all 4 combinations of transporter representations and molecule representations as the inputs for separate gradient boosted decision tree models [39]. Gradient boosting models consist of multiple decision trees that are constructed iteratively during training. In the first iteration, a single decision tree is built to predict the correct class for all pairs of transport proteins and potential substrates in the training data. By constructing new decision trees, subsequent iterations aim to minimize the errors made by the existing trees. Ultimately, an ensemble of diverse decision trees is formed, each focusing on different aspects of the input features and collectively striving to predict the correct outcome [39,40].

We performed extensive hyperparameter optimizations for all 4 models by performing 5-fold cross-validations (CVs) on the training set (for details, see the Methods subsection "Hyperparameter optimization of the gradient boosting models"). For each model, we selected the set of hyperparameters that resulted in the highest average Matthew's correlation coefficient (MCC) across all 5 folds of the CV. The MCC is a correlation coefficient for binary data, similar to Pearson's correlation coefficient for continuous data. An MCC of 0 would result from a model that randomly assigns data points to the positive and negative classes, while +1 would indicate perfect agreement. We chose the MCC as a measure of model performance because it is reliable even for imbalanced binary data sets [41].

After hyperparameter optimization, we trained each model with the best set of hyperparameters on the entire training set and validated model performance on the test set. In addition to the MCC, we used accuracy, ROC-AUC score, and precision as evaluation metrics. Accuracy is simply the proportion of correctly predicted data points among the test data. A model that randomly assigns positive and negative class labels would result in an accuracy of ~50%, whereas a model that always predicts the negative class would have an accuracy of 75%, a consequence of the unbalanced class distribution with 3 times as many negative compared to

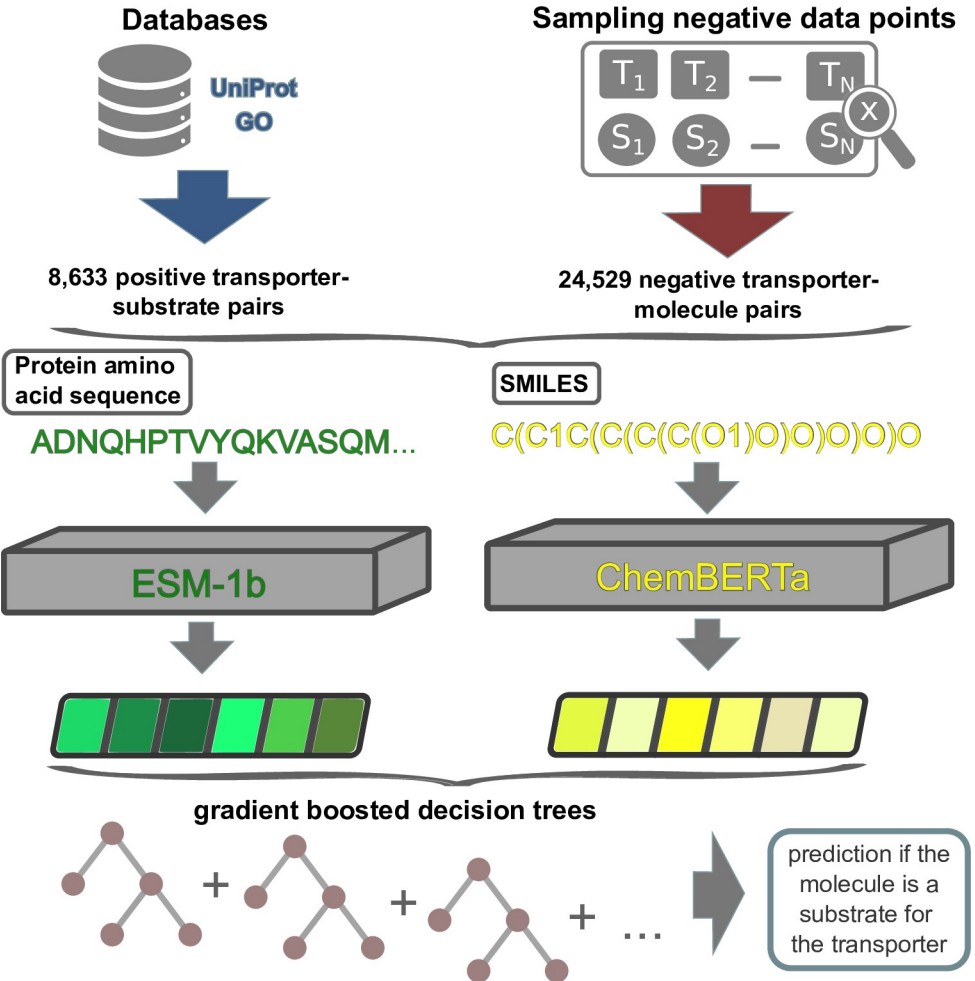

**Fig 1. Model overview.** Experimentally validated transporter-substrate pairs and sampled negative transporter-molecule pairs are numerically represented using the ESM-1b and ChemBERTa Transformer Networks. The concatenated transporter-molecule representations are used to train a gradient boosting model. After training, the fitted model can be used to predict whether a molecule is a substrate for a given transport protein.

positive data points. The ROC-AUC score is a value between 0 and 1 that summarizes how well a classifier is able to distinguish between the positive and negative classes, where any non-informative model would result in a value of 0.5, while a value of 1 would correspond to perfect predictions. The precision, also known as positive predictive value (PPV), is defined as the fraction of true positive predictions among all positive predictions.

Gradient boosted decision tree models with ESM-1b vectors as transporter representations outperformed models with ESM-1b$_{ts}$ vectors, and models with ChemBERTa vectors as molecule representations achieved better results than models with ECFP vectors (**Table 1** and

**Table 1. Prediction performance on the test set for all 4 combinations of protein and small molecule representations.**

|  | Accuracy | ROC-AUC | MCC | Precision |
|---|---|---|---|---|
| ESM-1b + ECFP | 91.5% | 0.956 | 0.78 | 0.84 |
| ESM-1b$_{ts}$ + ECFP | 90.0% | 0.955 | 0.75 | 0.80 |
| ESM-1b + ChemBERTa | 92.4% | 0.961 | 0.80 | 0.88 |
| ESM-1b$_{ts}$+ ChemBERTa | 90.6% | 0.957 | 0.76 | 0.82 |

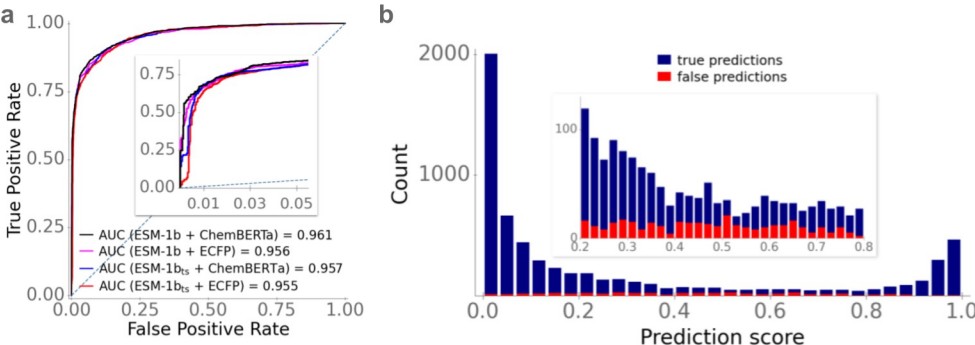

**Fig 2. Optimized gradient boosted decision tree models provide accurate predictions of transporter-substrate pairs. (a)** ROC curves for the test set for different combinations of transporter and molecule representations. The dotted line shows the ROC curve expected for a completely random model. **(b)** Stacked histogram bars show the prediction score distributions of true predictions (blue) and false predictions (red) for the best-performing model. The inset shows an enlargement of the interval [0.2,0.8]. The data underlying the graphs shown in this figure can be found in **S2 Data**.

**Fig 2A**). The best-performing model used the ESM-1b vector combined with the ChemBERTa vector as its input, achieving an accuracy of 92.4%, an ROC-AUC score of 0.96, an MCC of 0.80, and a precision of 0.88. We used McNemar's test to test if the difference in model performance between the 2 best models (ESM-1b + ECFP versus ESM-1b + ChemBERTa) is statistically significant. Indeed, the difference in model performance is statistically significant ($p = 0.003$). As a result, we chose ESM-1b vectors as protein representations and ChemBERTa vectors as molecule representations for the final SPOT model.

The SPOT model does not simply predict binary class labels, but instead outputs a score between 0 and 1 for each transporter-molecule pair, which can be interpreted as the probability that the data point belongs to the positive class. For our evaluation, we predicted a positive class label for all data points with a score >0.5 and the negative class label for all remaining data points. It is likely that the correct class label can be predicted with a much higher confidence if the prediction score is either close to 0 or close to 1. We plotted the distribution of predicted scores for all test data points in a histogram and visualized the proportion of correctly classified data points (**Fig 2B**). For 94.5% of all test data points, SPOT predicts scores that are either high (>0.6) or low (<0.4), achieving an accuracy of 94.2% for these data points. For the 5.5% of all test data points with a predicted score between 0.4 and 0.6, the accuracy is only 63.1% and hence indeed much lower. Thus, we conclude that the predicted score is a good indicator of how likely it is that the predicted class label is indeed true.

### SPOT performs well for different levels of sequence identity

Predicting the function of transport proteins is of particular interest, especially if no close homologs or proteins with highly similar amino acid sequence with known function exist. To investigate how good model performance is for those data points without any highly similar protein sequence in the training set, we calculated for all transporter-molecule pairs in the test set the maximal pairwise protein sequence identity compared to all protein sequences in the training set. On average, the maximum sequence identity of a test protein to any protein used for training is 62%. We calculated the MCC, the accuracy, and the ROC-AUC score for different levels of protein sequence identity (**Figs 3 and** S2A).

SPOT achieves a good model performance even for test data points with a low maximal sequence identity <40% compared to any sequence in the training set. For these 1,464 transporter-molecule pairs, the model achieves an accuracy of 84.0% and an MCC of 0.56,

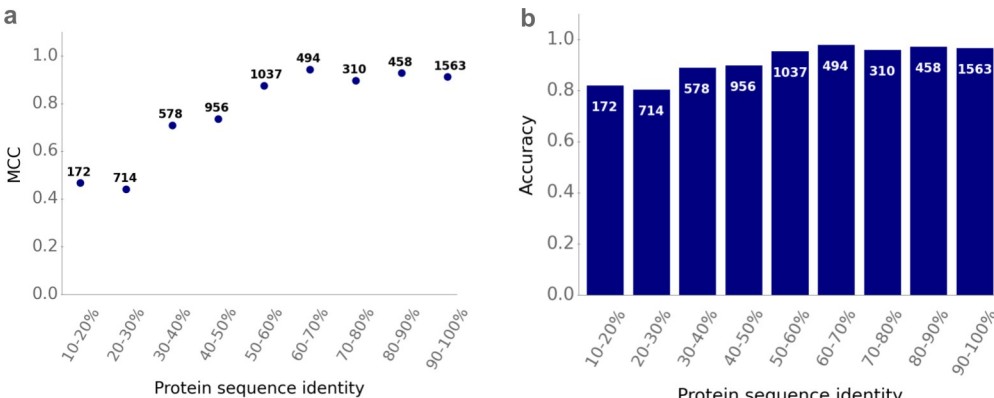

**Fig 3. Good prediction accuracies for different levels of protein sequence similarities.** We divided the test set into different subsets according to the maximal pairwise protein sequence similarities compared to all proteins in the training set. Panel **(a)** displays the MCC and panel **(b)** displays the accuracy for different levels of protein sequence similarity. The white numbers show the frequencies of test data points for each of the subsets. The data underlying the graphs shown in this figure can be found in **S2 Data**.

compared to an MCC of 0.00 that would result from a model that randomly assigns class labels. Thus, while model performance increases with increasing protein sequence identity (compared to sequences in the training set), we conclude that good predictions can be achieved even for those transporters that are only very distantly related to the proteins in the training set. For transport proteins with maximal sequence identity >50% compared to any sequence in the training set, the MCC and the accuracy remain approximately constant as the sequence identity increases further (**Fig 3**). The MCC plateaus at a value of ~0.90, while the accuracy plateaus at ~96%.

## SPOT performs well for substrates not present in the training data

We have shown above that model performance depends on how similar the protein sequences in the test set are compared to the proteins in the training set. Similarly, it is likely that model performance also depends on whether and how often a particular substrate of interest was already present in the training set. To investigate this, we calculated for each (potential) substrate in the test set, how often it occurred in the training set among all positive data points. We divided the test set into different subsets according to these frequencies, and we calculated the MCC, the accuracy, and the ROC-AUC score for each subset (**Figs 4 and** S2B).

While the experimental data for transport proteins is very limited and only 364 different substrates are included in our data sets, model performance is good even if the potential substrate did not occur in the training set. Thus, while the model is limited to seeing only a small number of different substrates during training, it learns to generalize and can be successfully applied to a large number of different substrates after training. Strikingly, the difference in model performance between test data points with substrates included and not included in the training set is only moderate. For the 166 test data points with unseen substrates, the accuracy is 86.7% and the MCC is 0.61. If the small molecule was present at least once in the training set, the model's performance increases to an accuracy of 92.6% and a MCC of 0.81.

## SPOT prediction performance depends on phylogenetic domain, subcellular localization, and transporter type

While the training and test data sets contain many transporters from eukaryotes and bacteria, transporters from archaea are underrepresented. To explore the potential implications of this

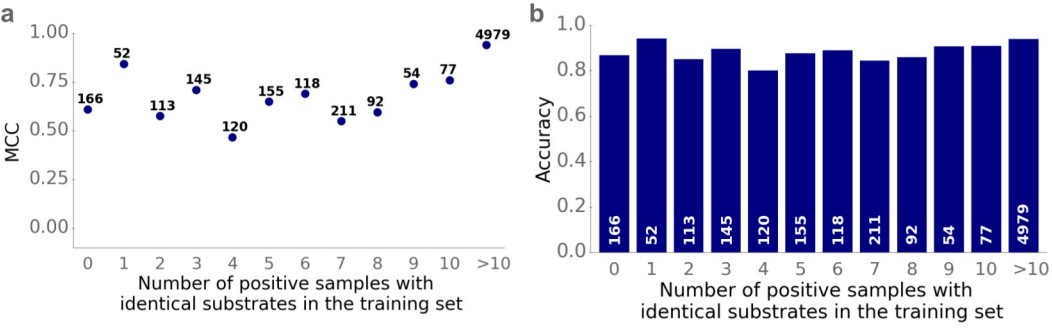

**Fig 4. SPOT provides accurate predictions for transport-molecule pairs, even if the molecule was not present in the training set.** We grouped small molecules by how often they occur as substrates among all positive data points in the training set. Panel **(a)** shows the MCC and panel **(b)** shows the accuracy for each group. The data underlying the graphs shown in this figure can be found in **S2 Data**.

data imbalance on model performance, we partitioned the test set into subsets corresponding to the 3 domains of life. We evaluated the performance of SPOT in each subset by calculating the accuracy, ROC-AUC score, and MCC (**Table 2**). Despite their underrepresentation in the training set, predictions for archaeal transporters are very good. Prediction performance was slightly weaker for eukaryotic proteins, despite the abundance of relevant training data.

To explore whether we could enhance model performance for eukaryotic transporters, we conducted another round of hyperparameter optimization for a gradient boosting model with a hyperparameter that reduces the influence of training data from prokaryotes. During CV, we selected the model with the highest mean MCC on the eukaryotic data points. We then trained this model on the entire training set and tested it on all eukaryotic test data points, resulting in an accuracy of 91.1%, an ROC-AUC score of 0.939, and an MCC of 0.763.

Thus, the refined model led to slightly improved predictions for eukaryotes, but still falls short of the general model's performance for prokaryotic transporters. One potential reason for this lower performance could be a higher amino acid sequence variability of eukaryotic transporters. However, the difference in performance between prokaryotes and eukaryotes cannot be explained by different distributions of maximum sequence identity between test proteins and training data (**S3 Fig**). Surprisingly, we found that for transporters with low sequence identity to training proteins, predictions were better for eukaryotes than for bacteria. We next hypothesized that prediction quality might differ between transporters in different subcellular localizations: while all prokaryotic transporters are positioned in 1 or 2 membranes located between the cytosol and the environment, many eukaryotic transporters facilitate transport between the cytosol and internal compartments. We obtained subcellular location information for the eukaryotic test data from UniProt, categorizing each data point into cell membrane; nuclear membrane; organelle membrane; or unknown/ambiguous. Prediction accuracy is indeed slightly lower for transport proteins located in organelle membranes compared to those located in the cell membrane (**Table 3**). However, predictions for transporters located in the eukaryotic cell membrane are still inferior to those for prokaryotic transporters,

**Table 2. Prediction quality for transporters partitioned by the domain of life of the source organism.**

| Domain | Fraction of test data | Fraction of training data | Accuracy | ROC-AUC | MCC |
|---|---|---|---|---|---|
| Bacteria | 43.3% | 48.4% | 95.3% | 0.985 | 0.875 |
| Eukarya | 53.5% | 50.2% | 89.6% | 0.934 | 0.727 |
| Archaea | 3.2% | 1.4% | 97.5% | 0.980 | 0.934 |

**Table 3. Prediction quality for eukaryotic transporters with different subcellular localizations.**

| Location | Fraction of test data | Accuracy | ROC-AUC | MCC |
|---|---|---|---|---|
| Cell membrane | 28.5% | 92.0% | 0.958 | 0.79 |
| Nucleus membrane | 0.9% | 96.6% | 0.981 | 0.906 |
| Organelle membrane | 21.1% | 90.2% | 0.935 | 0.735 |
| Unknown/ambiguous | 49.6% | 90.8% | 0.927 | 0.756 |

and thus subcellular localization can only partly explain the lower performance for eukaryotic transporters.

The transporters in our data set can be classified according to their mechanism of action: channels/pores (class 1); electrochemical potential-driven transporters (class 2); primary active transporters (class 3); and group translocators (class 4). SPOT demonstrates very high predictive accuracy for channels/pores (class 1), achieving an MCC of 0.80 (**S4 Fig**). Electrochemical potential-driven transporters (class 2) and primary active transporters (class 3) also show good model performance, with MCCs surpassing 0.61. Group translocators (class 4) showed slightly worse model performance, with an MCC of 0.54. Notably, there are significantly fewer data points assigned to class 4 compared to all other classes (**S4 Fig**), and thus the slight decrease in model performance might be related to an underrepresentation of group translocators in the training set.

## SPOT outperforms previous models for predicting substrate classes of transport proteins

Because SPOT is the first general model for predicting transporter-substrate pairs, we had no existing benchmark against which to compare its performance. To nevertheless compare the performance of our approach to that of previous methods, we also developed a related model for a similar, less specific task: predicting substrate classes for transport proteins using only information extracted from the transporter amino acid sequence [16–19]. A simple approach for predicting the substrate class of a given transporter would be to apply SPOT to numerous substrates from different substrate classes. However, this approach would likely result in many false positive predictions. We thus trained a new model that directly predicts substrate classes from the protein amino acid sequences.

A recent approach called TooT-SC [19] predicted substrate classes for transport proteins by mapping each transporter to one out of 11 different substrate classes. Most previous approaches instead predicted 7 different substrate classes: cations, anions, sugars, amino acids/oligopeptides, proteins/mRNA, electrons, and other substrates [16–18]. To compare our model with these previous methods, we mapped the transporters in our data set to one of the 7 substrate classes (for more details, see "Obtaining training and validation data sets", Results). This resulted in a data set of 11,664 data points, which we divided into 80% training data and 20% test data as described above.

To obtain numerical representations for all proteins in our data set, we computed the ESM-1b vectors. These vectors served as the input to a gradient boosted decision tree model, which we trained to predict the transporters' substrate classes. We performed a 5-fold CV on the training set to identify the optimal set of hyperparameters. Our goal was to select hyperparameters that would produce a prediction model capable of achieving high accuracies even for proteins with low similarity to any protein in the training set. To achieve this, for the 5-fold CV we again divided the training set into 5 folds in such a way that the pairwise sequence similarity between protein sequences from different folds would not exceed 60%. In addition, we

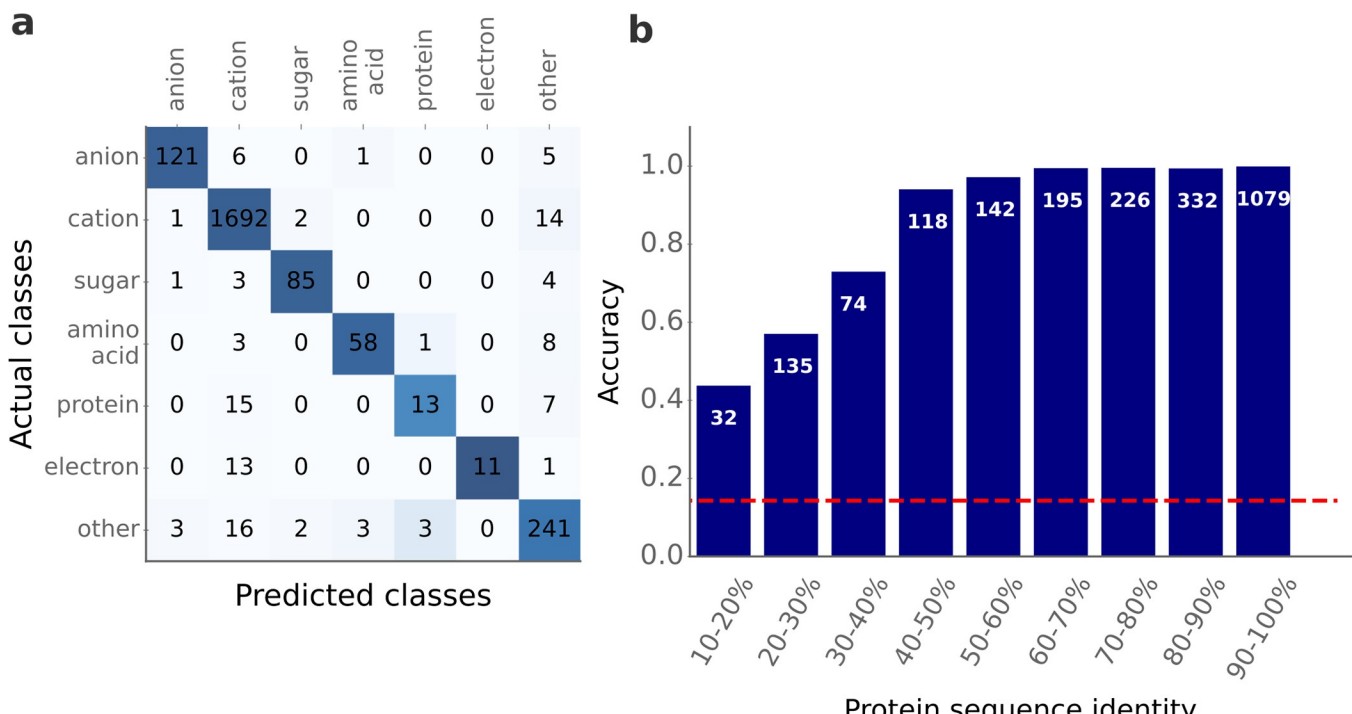

**Fig 5. Good prediction accuracies for different substrate classes and different levels of protein sequence identity. (a)** Confusion matrix for all 7 substrate classes. Actual classes are arranged vertically and predicted classes are arranged horizontally. Darker colors indicate higher numbers; the colors of each column are normalized by the number of total actual samples for that class. **(b)** We divided the test set into different subsets according to the maximal pairwise protein sequence similarity compared to all proteins in the training set (displayed on the x-axis). Each bar indicates the mean accuracy for the corresponding subset. The white numbers show the number of test samples in each of the subsets. The red dashed line indicates the accuracy of a prediction model that assigns class labels randomly. The data underlying the graphs shown in this figure can be found in **S2 Data**.

incorporated a weight parameter to address the unbalanced number of class labels (**S1 Table**), which reduced the influence of data points originating from the most common class.

After performing the 5-fold CV, the set of hyperparameters that yielded the highest mean accuracy across all folds was selected to train a gradient boosted decision tree model on the entire training set. The resulting model achieves an accuracy of 95.1% on the independent test data set of 2,333 data points. The model achieves high accuracies for all 7 substrate classes (**Fig 5A** and **S2 Table**) and outperforms the previous state-of-the-art method, TranCEP, which achieved an overall accuracy of 74.2% on independent test data. While it is not possible to make a completely fair comparison between the 2 models, which were trained and evaluated on different data, the large difference in model performance indicates that the SPOT model architecture and the newly generated data set lead to superior prediction quality. We note that the "cation" and "other substrates" classes have a higher number of incorrect transporter predictions compared to all other classes (**Fig 5A**), probably due to their overrepresentation in the training and the test set (**S1 Table**).

Analogous to the binary classification task, prediction accuracy is good even for transporters without highly similar protein sequences in the training set (**Fig 5B**). For the 241 test data points with a maximal pairwise sequence identity <40% compared to any sequence in the training set, the average prediction accuracy is 60.2%. To put this in perspective, the naive approach of randomly assigning class labels to all test data points would result in an average accuracy of 14.3%.

## SPOT outperforms simpler baseline approaches

To justify the use of a sophisticated machine learning model, its performance should be compared to a much simpler baseline model. Therefore, in addition to benchmarking SPOT against previously published models, we also explored a much simpler, similarity-based approach for the prediction of transporter-substrate pairs. For a given target protein, this approach involved identifying the $k$ most similar proteins with experimentally validated substrates in our training database, with $k$ a model hyperparameter. The model then predicts all molecules associated with the most similar proteins as positive substrates of the target protein. To compare the similarity-based approach to SPOT, for each test sequence we classified all substrates identified by the similarity approach that are among the experimentally validated substrates of the test sequence as correct predictions. We classified all true positive substrates of the test sequence that could not be identified by the similarity approach as incorrect predictions, and we further classified those substrates that were identified by the similarity approach but were not among the positive substrates of the test sequence as incorrect predictions. This approach may lead to a slightly overly pessimistic estimate of the false positive rate, but as this scoring is applied to both approaches for their comparison, the test remains fair.

We compared the accuracy, recall, and precision between SPOT and this similarity-based approach for different values for k, the number of most similar proteins considered ($k$ = 1,2,3,4; **Table 4**). At $k$ = 1, the similarity-based approach's accuracy (64.2%) is much lower than that of SPOT (92.5%), decreasing further with increasing $k$. While at higher values of $k$, the similarity-based approach can achieve high recall—correctly identifying a high fraction of true positive substrates—it does so at the cost of very low precision, i.e., it predicts a high number of false positives. In contrast, SPOT achieves high precision and recall simultaneously.

Our test set comprises 1,637 positive transporter-substrate pairs. For $k$ = 1, the similarity-based approach predicted 1,587 positive pairs, of which 1,252 are experimentally validated (true positives). This means that 385 positive substrates from the test set were not identified, corresponding to a recall of 76.5% for the similarity-based approach. In comparison, SPOT achieves a higher recall of 83.1%.

As it is more likely that 2 proteins share the same set of substrates if they have a high sequence similarity, we separately analyzed the recall for transporters without similar protein sequences in the training set. For transporters with a maximal sequence identity of under 30% compared to sequences in the training set, the similarity-based approach with $k$ = 1 yielded a recall of only 23%. In contrast, SPOT achieves a significantly higher recall rate of 43% for these proteins (**S5 Fig**). We thus conclude that SPOT is a valuable tool, especially when there are no close homologs with known function. Additionally, while SPOT leads to higher recall rates, it can also determine the probability that a molecule is not a substrate for a given transport protein, providing further useful information compared to the similarity-based approach.

We also compared SPOT with another simple approach, where the candidate substrates are one-hot encoded, instead of using the substrate encodings obtained from the ChemBERTa

**Table 4. Comparison of SPOT to a similarity-based approach.**

| Method | Accuracy | Precision | Recall |
|---|---|---|---|
| SPOT | **92.5%** | **88.0%** | **83.1%** |
| Similarity approach ($k$ = 1) | 64.2% | 78.9% | 76.5% |
| Similarity approach ($k$ = 2) | 58.8% | 68.1% | 80.0% |
| Similarity approach ($k$ = 3) | 50.5% | 56.8% | 80.9% |
| Similarity approach ($k$ = 4) | 44.9% | 49.5% | 81.8% |

transformer network. This approach has the disadvantage that it can only be applied to substrates that were present in the training set. However, a one-hot encoding-based approach allows us to estimate how much SPOT learns from the molecular information encoded in the ChemBERTa vectors as opposed to just learning the substrate's identity. For test data points with substrates that were present in the training set, this approach results in a lower accuracy of 88.2%, MCC of 0.72, and ROC-AUC score of 0.950 compared to SPOT, which achieves an accuracy of 92.6%, an MCC of 0.806, and an ROC-AUC score of 0.963 for these data points.

## Comparison to experimental results for screening candidate substrates

To validate the performance of SPOT on experimentally identified positive and negative data points, we used data from a high-throughput screening method for identifying candidate substrates of transporters. Majd and colleagues [42] identified candidate substrates for 2 transporters, GalP and AAC, with different structures and transport mechanisms. GalP is a member of the sugar porter family, while AAC belongs to the mitochondrial carrier family. GalP and AAC were each tested against a library of 30 suitable, distinct compounds. As neither GalP nor AAC were part of the SPOT training set, this data provides a valid test case. The protein sequences of GalP and AAC have a maximum sequence similarity of 56% and 58%, respectively, compared to all training sequences. We were able to map all tested compounds to InChI strings, with the exception of 1 potential substrate for the GalP transporter and 3 potential substrates for the AAC transporter. It is worth noting that only 9 out of the remaining 56 compounds were present in our training set.

Using the previously trained SPOT model, we made predictions for all mapped transporter-substrate pairs in the Majd and colleagues data set [42]. SPOT achieves an accuracy of 86.2% and an MCC of 0.53 for the GalP transporter. The model correctly identifies 2 out of 6 experimentally identified candidate substrates, while correctly classifying the remaining 23 compounds as non-substrates. Similarly, when applied to the AAC transporter, SPOT achieves an accuracy of 81.5% and an MCC of 0.48. The model identifies 2 out of 7 candidate substrates and accurately classifies all 20 non-substrates. The reason for the high accuracy, despite the limited recall, is that SPOT correctly identified all non-binding molecules, which make up the majority of the test data. It is important to note that the screening method employed by Majd and colleagues [42] identifies candidate substrates, but does not guarantee that the identified compounds are indeed true substrates for the transporters. This could explain why our model only correctly predicted around one third of the experimentally identified candidate substrates as true substrates.

## Web server and python functions allow an easy use of prediction models

To simplify the use of our developed prediction models, we have implemented a web server and a Python function. The web server, accessible at https://spot.cs.hhu.de, enables the utilization of our prediction tool through a web browser without requiring the installation of any additional software. We also developed a Python function that is available at https://github.com/AlexanderKroll/SPOT. For these prediction models, we trained gradient boosted decision tree models on both the training and test sets to expand the amount of training data, and thus improve the model's performance.

## Discussion

Here, we present the first general approach for predicting specific substrates for transport proteins. SPOT achieves an accuracy of over 92% on an independent test set. Notably, SPOT performs well even when dealing with transporters that have a low sequence identity of <40%

compared to the proteins in the training set, achieving an accuracy of approximately 84% and an MCC of 0.56. We achieved these results by using very general input features: to numerically represent the transport proteins and the potential substrates, we used 2 Transformer Networks trained to process protein amino acid sequences and SMILES strings of small molecules, respectively. These models are currently the state-of-the-art for encoding protein and small molecule information.

In a previous study of enzyme-substrate pair prediction, we found that using task-specific enzyme representations significantly improved prediction accuracy [31]. However, we did not find such an improvement when incorporating task-specific transporter representations in the present study, possibly due to the scarcity of available training data points. The ESM-1b model, which contains over 670 million parameters, requires a significant amount of training data for effective parameter tuning. For comparison, when creating task-specific enzyme representations for predicting enzyme-substrate pairs, we had access to 287,386 positive training data points; in contrast, only 8,633 positive transporter-substrate pairs were available for fine-tuning the ESM-1b model here.

Instead of using the linear amino acid sequence for the generation of protein representations, one could alternatively use the 3D protein structures for functional predictions. This approach seems preferable in principle, since the function of a protein is closely linked to its structure. Recent advances, with models such as AlphaFold [7], ESMFold [8], and RoseTTA-Fold [9], have revolutionized the prediction of 3D protein structures from amino acid sequences. Even for transmembrane proteins, which are typically underrepresented among proteins with experimental 3D structures, the corresponding structure predictions are highly accurate [43]. Unfortunately, current techniques for extracting protein representations from predicted structures do not surpass the performance of protein representations extracted directly from the proteins' amino acid sequence for functional predictions [24]. However, once methods emerge that effectively extract the relevant information from protein 3D structures, their utilization will likely improve model performance.

One of the major challenges in predicting transporter-substrate relationships is the lack of experimentally confirmed non-substrates for transport proteins. To overcome this obstacle, we developed a carefully designed strategy to randomly sample negative transporter-molecule pairs. While this data augmentation technique may lead to false negative data points, such false negatives are a priori expected to be rare, an expectation confirmed a posteriori by the good results on independent test data. However, not only the absence of negative data, but also the quantity and quality of positive data points impose constraints. We observed that model performance increases with increasing protein sequence similarity and when potential substrates are included in the training set. Consequently, model performance for new transporter-molecule pairs is expected to improve as more experimentally validated training data become available.

Since the different databases used to construct our data set were compiled using different strategies, we wondered if there is a significant difference in model performance between test data derived from the UniProt and GO databases. We divided the test data set into 2 corresponding splits and further divided these 2 sets into subsets according to the maximum protein sequence identity compared to the training proteins. The results show that the model performance is substantially better for test data points extracted from UniProt (S6 Fig). We hypothesize that the reason for the lower performance on test data points from GO is lower data quality, i.e., the GO database may contain more incorrect transporter-substrate pairs compared to UniProt. If this is indeed the case, the true model performance of SPOT is best approximated by the performance on test data points downloaded from UniProt, and thus the true model performance may be even higher than reported above.

The experimental characterization of substrates for transport proteins is often challenging and expensive [42]. With a true positive rate of approximately 82% and a false positive rate of approximately 11%, SPOT is likely to yield satisfactory results when applied to individual transport proteins and a preselected set of potential substrates. For example, if 50 molecules are selected as candidate substrates, and one of these 50 molecules is the true substrate, an FPR of 11% means that the model will only incorrectly predict about 5 molecules as substrates, with an 82% chance of correctly identifying the true substrate. On the other hand, if SPOT is applied to a large set of transport proteins, each of which is paired with many potential substrates, a false positive rate of 11% is likely to result in too many false predictions. We conclude that the most promising use case is the study of single transporter proteins, where SPOT can select the most likely substrates from a set of candidate molecules, thereby reducing the subsequent experimental burden to a manageable level.

## Methods

### Software and code availability

All software was written in Python [44]. We generated protein representations using the deep learning library PyTorch [45]. We fitted the gradient boosting models using the XGBoost library [39]. We used the Django web framework [46] to implement the SPOT web server. The code used to generate the results of this paper, in the form of Jupyter notebooks, is available at https://github.com/AlexanderKroll/SPOT and at https://doi.org/10.5281/zenodo.13234753. All data sets used to generate the results of this manuscript are available at https://doi.org/10.5281/zenodo.8358682.

### Preprocessing GO terms

Entries in the GO Annotation database [25] are associated with GO terms that contain information about the biological processes, molecular functions, or cellular components of gene products [47]. Before searching all entries in the GO annotation database, we checked all GO terms for information about transport proteins and generated a list of all 2,559 such GO terms that specify the function of transporters. Next, we tried to extract the name of the transported substrate and we mapped the substrate names to a ChEBI ID [48] using the KEGG [49] and PubChem [50] databases. We discarded all GO terms for which we could not find a ChEBI ID. This resulted in a data set of 808 GO terms with ChEBI IDs for the substrates.

### Extracting transporter-substrate pairs from the GO annotation database

The GO Annotation database for UniProt IDs contains over 922 million entries that associate UniProt IDs, which are identifiers for proteins, with GO terms. We extracted all ~44 million entries with GO terms containing information about transport proteins. Entries in the GO annotation database have different levels of evidence: experimental, phylogenetically inferred, computational analysis, author statement, curator statement, and electronic evidence. The vast majority of annotations have only electronic evidence, i.e., they are not manually curated. We excluded all data points without experimental evidence, resulting in a data set with 12,913 pairs of UniProt IDs and GO terms.

We mapped all GO terms to the transported substrates and kept only data points with ChEBI identifiers for the substrates. We used the UniProt mapping service to map the UniProt IDs to their corresponding amino acid sequences [2]. The resulting data set consisted of 4,775 data points with 3,420 different UniProt IDs and 281 different ChEBI IDs.

### Extracting transporter-substrate pairs from the UniProt database

We downloaded all 567,483 reviewed protein entries from the UniProt database [2]. We extracted RHEA reaction IDs [51] for 17,510 entries with transport reactions, and we extracted ChEBI IDs for all substrates using the RHEA reaction IDs. We used the UniProt mapping service to map the UniProt IDs to their corresponding amino acid sequences [2]. The resulting data set consisted of 15,864 data points with 12,910 different UniProt IDs and 243 different ChEBI IDs.

### Extracting transporter-substrate pairs from the Transporter Classification Database

The TCDB contains 21,938 transport proteins that are linked to 1,701 transport families by TC numbers [26]. TC numbers classify transporters into different transport families that contain information about their function. We used a file from the TCDB to link TC numbers to specific substrates via ChEBI IDs. The resulting data set consisted of 18,051 data points with 11 799 different UniProt IDs and 1,559 different ChEBI IDs.

### Merging data sets with transporter-substrate pairs

We merged the transporter-substrate pair data sets obtained from the GO Annotation database and the UniProt database. After removing 2,633 duplicate data points that occurred in more than 1 database, the resulting data set consists of 18,006 data points with 13,777 different UniProt IDs and 509 different ChEBI IDs.

### Creating a data set for the transporter-substrate pair prediction

We attempted to map all 509 ChEBI IDs in the final data set to an InChI string containing structural information about the molecules; 83 ChEBI IDs could not be mapped to InChI strings, e.g., because some ChEBI IDs refer to classes of molecules such as sugars, ions, or proteins rather than to specific molecules. We removed 17 proteins with sequence lengths less than 30 amino acids to avoid including fragments of proteins. Additionally, we removed all ~7,500 transporter-substrate pairs with a proton as a substrate, because we hypothesized that for the majority of these data points, the transport protein is either a symporter or an antiporter with the proton not being the primary substrate. This resulted in a data set of 8,587 data points comprising 5,882 unique transport proteins and 364 different substrates.

We divided this data set into 80% training data and 20% test data in such a way that no identical transporter sequences occur in both the training and test set. For a detailed analysis of the predictive capabilities of our model, we calculated the maximal pairwise sequence identity for each test sequence compared to all sequences in the training set using the Needleman–Wunsch algorithm from the EMBOSS software package [29]. To perform a 5-fold CV for hyperparameter optimization of the machine learning models, we split the training set into 5 folds. We wanted to obtain a set of hyperparameters that would lead to a prediction model that generalizes well to proteins that are not highly similar to proteins in the training data. Thus, we generated the 5 folds such that 2 proteins from 2 different folds never have a pairwise sequence identity of >60%. To achieve this, we used the cd-hit algorithm [52], which clusters protein sequences according to their sequence similarity.

### Sampling negative data points

For each positive transporter-substrate pair in our data set, we generated between 1 and 4 negative data points for the same transport protein by randomly sampling small molecules as non-substrates. On average, we obtained 3 negative data points for each positive transporter-

substrate pair. Distinguishing between true and false substrates is more difficult for molecules that are similar to the true, known substrates. To challenge our model to learn this distinction, we restricted our sampling of negative data points to molecules similar to the true substrate. For this purpose, we first computed the pairwise similarity of all substrates in our data set using the FingerprintSimilarity function from the RDKit package DataStructs [53]. This function takes the molecular fingerprints of the molecules as input and computes values between zero (no similarity) and one (high similarity). When possible, we sampled molecules with similarity scores between 0.75 and 0.95. If we did not find such molecules, we reduced the lower bound in steps of 0.2 until enough small molecules could be sampled. Since 2 similar transport proteins are more likely to share the same set of substrates, when sampling non-substrates for a given transporter, we excluded all molecules that occurred as a true substrate for any transport protein in our data set with a sequence similarity >60%. This reduces the chance of sampling false negative data points. During this sampling process, we took the distribution of molecules among the positive data points into account, i.e., molecules that occur more frequently as substrates among the positive data points also occur more frequently among the negative data points. To achieve this, we excluded molecules from the sampling process if these molecules were already sampled enough times (i.e., 3 times their total occurrence in the set of positive transporter-substrate pairs).

## Generating a data set for the substrate class prediction

To generate a data set for the substrate class prediction task, we mapped all 509 ChEBI IDs in our data set to one of the following categories: cations, anions, sugars, amino acids/oligopeptides, proteins/mRNA, electrons, and other substrates. We used the ChEBI IDs and the Python package bioservices [54] to download information about the molecules from ChEBI [48]. We downloaded chemical formulas, InChI strings [27], molecule names, and information about the parent ChEBI IDs. When possible, we used the type of the parent ChEBI ID to classify the molecules. In addition, we classified molecules with names ending in "ose" and containing only C, H, and O atoms as sugars if they were not already classified as sugars.

Molecules that could not be classified into one of the first 6 categories (cations, anions, sugars, amino acids/oligopeptides, proteins/mRNA, and electrons) were classified as other molecules. To avoid misclassification, we only considered molecules for this category if we were able to download the substrate name and chemical formula. We then manually checked the assignment of ChEBI IDs to the 7 classes to find and correct misclassifications. In total, we were able to assign 423 of the 509 ChEBI IDs to one of the 7 substrate classes.

We mapped these substrate classes to our data set of transporter-substrate pairs. If we found multiple or no substrate classes for a transporter amino acid sequence, we discarded that data point. This resulted in a data set of 11,679 transport proteins. We removed 15 proteins with sequence lengths less than 30 amino acids to avoid including protein fragments. We computed the ESM-1b vector for each transporter in the data set as a protein representation [21] (see below).

We split the data set into 80% training data and 20% test data so that no identical transporter sequences occur in both the training and test set. For a detailed analysis of the predictive capabilities of our model, we calculated the maximum pairwise sequence identity for each test sequence compared to all sequences in the training set using the Needleman–Wunsch algorithm from the EMBOSS software package [29]. To perform a 5-fold CV for hyperparameter optimization of the machine learning models, we split the training set into 5 folds. We wanted to obtain a set of hyperparameters that would lead to a prediction model that generalizes well to proteins that are not highly similar to proteins in the training data. Thus, we generated the 5

folds in such a way that 2 proteins from 2 different folds would not have a pairwise sequence similarity greater than 60%. To achieve this, we used the cd-hit algorithm [52], which clusters protein sequences according to their sequence similarity.

## Computing transporter representations

We used the ESM-1b model [21] to compute 1,280-dimensional numerical representations for all transport proteins. The ESM-1b model is a Transformer Network [20] that takes amino acid sequences as input and produces numerical representations of the sequences. First, each amino acid within a sequence is converted into a 1,280-dimensional representation that encodes both the type of amino acid and its position in the sequence. These representations are then iteratively updated for 33 update steps using information about the representation itself as well as about all other representations of the sequence using the attention mechanism [55]. The attention mechanism allows the model to selectively focus on only relevant amino acid representations for updates [55]. During training, approximately 15% of the amino acids in a sequence are randomly masked, and the model is trained to predict the identities of the masked amino acids. The ESM-1b model was trained on approximately 27 million proteins from the UniRef50 data set [36]. To generate a single representation for the entire transporter, ESM-1b computes the element-wise mean of all updated amino acid representations in a sequence [21]. We generated these representations for all transporters in our data set using the code and trained ESM-1b model provided by the Facebook AI Research team on GitHub.

## Computing task-specific transporter representations

To generate task-specific transporter representations for our task of predicting transporter-substrate pairs, we modified the ESM-1b model. For each input sequence, in addition to the representations of all individual amino acids, we added a token representing the entire protein. This protein representation is updated in the same way as the amino acid representations. The parameters of this modified model are initialized with the parameters of the trained ESM-1b model. After the last update layer of the model, i.e., after 33 update steps, we take the 1,280-dimensional representation of the whole transporter and concatenate it with a representation for a molecule, the 1,024-dimensional ECFP vector.

This concatenated vector is then used as input to a fully connected neural network (FCNN) with 2 hidden layers of size 256 and 32. The entire model was trained end-to-end for the binary classification task of predicting whether the added molecule is a substrate for the given transporter. This training procedure challenged the model to store all necessary transporter information for the prediction task in the transporter representation. After training the modified model, we extracted the updated and task-specific representation, the ESM-1b$_{ts}$ vectors, for each transporter in our data set.

We implemented and trained this model using the Python package PyTorch [45]. We trained the model on our data set of 33,162 transporter-molecule pairs for 30 epochs on 4 NVIDA DGX A100s, each with 40 GB of RAM. Training the model for more epochs did not improve the results. Due to the immense computing power and long training times, it was not possible to perform systematic hyperparameter optimization. We chose the hyperparameters after trying a few selected hyperparameter settings with values similar to those used to train the original ESM-1b model.

## Computing substrate representations

To generate learned substrate representations, we first mapped all (potential) substrates to SMILES strings, which are string representations that capture the molecular structure of small

molecules. We used these SMILES strings as input for the ChemBERTa model, which is a Transformer Network for processing SMILES strings [22]. ChemBERTa first decomposes the SMILES strings into disjoint tokens, then maps each token to a token embedding, and finally iteratively updates each token embedding for 3 steps. ChemBERTa was trained on ~77 million different SMILES strings. We applied this trained model to every SMILES string in our data set and extracted a learned 767-dimensional embedding from the last layer of ChemBERTa by computing the element-wise mean of all updated token embeddings.

## Hyperparameter optimization of the gradient boosting models

For the hyperparameter optimizations of all gradient boosting models, we performed 5-fold CVs by dividing the training set into 5 disjoint subsets (folds) of approximately same sizes. We aimed to select hyperparameters that lead to models that generalize well even to transporters that are not highly similar to proteins already present in the training set. To achieve this, we split the training set in such a way into 5 folds that 2 proteins from 2 different folds would never have a pairwise protein sequence identity above 60%. To achieve this, we used the cd-hit algorithm [52], which clusters protein sequences according to their sequence similarity. The threshold of 60% is somewhat arbitrary. However, we hypothesized that the exact value of this threshold is not decisive for the final model performance, and since the training process is already computationally expensive, we decided to not treat the threshold as an additional hyperparameter.

In each of the 5 runs of a CV, we aimed to predict the data points in one of the folds, using the remaining 80% of the data for training. We used the Python package hyperopt [56] to perform a random grid search for the following hyperparameters: learning rate, maximum tree depth, lambda and alpha coefficients for regularization, maximum delta step, minimum child weight, and number of training epochs. For the task of predicting transporter-substrate pairs, we added a weight parameter for the negative data points to account for the imbalance in class labels; this parameter allows the model to assign a lower weight to the negative data points during training. After hyperparameter optimization, we selected the set of hyperparameters with the highest mean MCC during CV. We used the Python package xgboost [39] to train the gradient boosting models.

## Determining the subcellular location of transporters

To determine the subcellular location of eukaryotic transport proteins, we downloaded subcellular location information for each protein from UniProt using the protein's UniProt ID. Based on this information, we attempted to categorize each protein into one of the following locations: cell membrane, organelle membrane, and nucleus membrane. Those proteins that either contained multiple locations or no specific location were classified as "unknown/ambiguous."

## Fitting a eukaryote-specific gradient boosting model

To fit a model that leads to high performance for data points from the Eukarya domain, we performed another round of hyperparameter optimization for a gradient boosting model with an additional weight parameter that reduces the influence of training data from prokaryotes. This weight parameter is a value between 0 and 1 and was determined during hyperparameter optimization. During CV, we trained the models on data from all domains, but validated the models only on eukaryotic data. We selected the model with the highest mean MCC during CV. After CV, we trained the model on the entire training set, but with less influence from the data points outside the Eukarya domain according to the weight parameter. This model was then validated on all test data points from eukaryotes.

## Classifying test data points according to transporter classes

We classified the transporters in our test data set according to their mechanism of action into channels/pores (class 1), electrochemical potential-driven transporters (class 2), primary active transporters (class 3), and group translocators (class 4). To assign these classifications, we utilized TC classes [26] obtained from UniProt whenever available, resulting in the successful assignment of 1,634 out of 6,282 test data points to one of these 4 categories.

## Preparing data from experimental results for screening candidate substrate

Majd and colleagues [42] developed a high-throughput screening method for identifying candidate substrates of transporters. Their method is based on the principle that transport proteins recognize substrates through specific interactions, leading to increased stabilization of the transporter population. Majd and colleagues [42] measured the apparent melting temperature Tm, the temperature at which the unfolding rate of a given population is highest, for transport proteins in the presence of potential substrates. An increase in the apparent melting temperature indicates an overall improvement in stability, suggesting the compounds as potential substrates. In their study, Majd and colleagues [42] applied this method to test the substrate specificity of 2 different transporters, GalP and AAC, for each utilizing a library of 30 different compounds. For our analysis, we considered compounds to be substrates if they resulted in an average increase in apparent melting temperature of at least 1 degree Celsius compared to the protein without any added compound.

To integrate the experimental results into our analysis, we attempted to map all tested compounds to InChI strings using the PubChem database [50]. One tested compound for the GalP transporter and 3 tested compounds for the AAC transporter could not be mapped to InChI strings.

## Supporting information

**S1 Data. Data set in xlsx-format containing all substrates in our data set, their number of occurrences, and their substrate class.**
(XLSX)

**S2 Data. Data set in xlsx-format containing the underlying data of all figures that are displaying data and results.**
(XLSX)

**S1 Text. Computing task-specific protein representations.**
(DOCX)

**S1 Table. The number of data points in each of the 7 substrate classes for the training and test set, and the number of different substrate ChEBI IDs that fall into each category.**
(TIFF)

**S2 Table. Accuracy, MCC, and ROC-AUC for each of the 7 classes compared to all other classes.** To calculate these metrics, each class was considered as the positive class, while all other classes were treated as the negative class.
(TIFF)

**S1 Fig. A majority of TCDB transport protein annotations appear not to be based on strong evidence.** UniProt IDs extracted from the TCDB are grouped by their annotation scores in the UniProt database. The annotation scores range from 1 to 5, where higher values indicate better evidence for the protein annotation. Direct experimental evidence corresponds

to a score of 5. The data underlying the graph shown in this figure can be found in **S2 Data**.
(TIF)

**S2 Fig. ROC-AUC score increases for increased protein sequence identity but not for molecules present in the training set. (a)** We divided the test set into different subsets according to the maximal pairwise protein sequence similarities compared to all proteins in the training set. We calculated the ROC-AUC score for each subset. **(b)** We grouped small molecules by how often they occur as substrates among all positive data points in the training set. We calculated the ROC-AUC score for each group. The data underlying the graphs shown in this figure can be found in **S2 Data**.
(TIF)

**S3 Fig. SPOT performs better for prokaryotic transporters than for eukaryotic transporters.** We divided the test set into 3 classes based on the domain of the source organism of the transport protein: Eukaryota, Bacteria, and Archaea. We further divided each class into subsets according to the maximum sequence identity compared to the training proteins. The plot shows the Matthew's correlation coefficient (MCC). The areas of the circles are proportional to the number of data points in each subset. The data underlying the graph shown in this figure can be found in **S2 Data**.
(TIF)

**S4 Fig. SPOT performs well across different classes of transport proteins.** We partitioned the test set into 4 distinct classes based on transport mechanisms annotated in TCDB: channels/pores (class 1); electrochemical potential-driven transporters (class 2); primary active transporters (class 3); and group translocators (class 4). The plot shows the Matthew's correlation coefficient (MCCs) for data points from each class. The areas of the circles are proportional to the number of test data points in each subset. The data underlying the graph shown in this figure can be found in **S2 Data**.
(TIF)

**S5 Fig. Improved recall compared to a similarity-based approach especially for proteins dissimilar to training data.** We divided the test set into subsets based on the maximum pairwise protein sequence similarities compared to all proteins in the training set. For each subset, we computed the recall of SPOT and a simpler, similarity-based method that considered the substrates of the most similar protein from the training set. The data underlying the graph shown in this figure can be found in **S2 Data**.
(TIF)

**S6 Fig. Data points extracted from UniProt can be predicted more accurately.** We partitioned the test set into 2 subsets based on their origin—UniProt or GO. The plot shows the Matthew's correlation coefficient (MCC) for different levels of maximal sequence identity compared to training proteins. The areas of the circles are proportional to the number of test data points in each subset. The data underlying the graph shown in this figure can be found in **S2 Data**.
(TIF)

## Acknowledgments

We thank Peter Schubert and Lutz Schmitt for helpful discussions. Computational support and infrastructure was provided by the "Centre for Information and Media Technology" (ZIM) at the University of Düsseldorf (Germany).

## Author Contributions

**Conceptualization:** Alexander Kroll, Martin J. Lercher.

**Data curation:** Alexander Kroll, Nico Niebuhr.

**Formal analysis:** Alexander Kroll, Gregory Butler, Martin J. Lercher.

**Funding acquisition:** Martin J. Lercher.

**Investigation:** Alexander Kroll, Martin J. Lercher.

**Methodology:** Alexander Kroll, Nico Niebuhr, Martin J. Lercher.

**Project administration:** Martin J. Lercher.

**Resources:** Martin J. Lercher.

**Software:** Alexander Kroll.

**Supervision:** Martin J. Lercher.

**Validation:** Alexander Kroll.

**Visualization:** Alexander Kroll.

**Writing – original draft:** Alexander Kroll, Martin J. Lercher.

**Writing – review & editing:** Alexander Kroll, Gregory Butler, Martin J. Lercher.

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
