## [Editor Report · Decision Letter 0]

18 Jun 2024

Dear Dr Lercher, 

Thank you for submitting your manuscript entitled "A general prediction model for substrates of transport proteins" for consideration as a Research Article by PLOS Biology.

Your manuscript has now been evaluated by the PLOS Biology editorial staff and I am writing to let you know that we would like to send your submission out for external peer review.

Once your full submission is complete, your paper will undergo a series of checks in preparation for peer review. After your manuscript has passed the checks it will be sent out for review. To provide the metadata for your submission, please Login to Editorial Manager (https://www.editorialmanager.com/pbiology) within two working days, i.e. by Jun 20 2024 11:59PM.

Kind regards,

Suzanne

Suzanne De Bruijn, PhD, 

Associate Editor

PLOS Biology

sbruijn@plos.org

---

## [Decision Letter · Decision Letter 1]

31 Jul 2024

Dear Dr Lercher,

Thank you for your patience while we considered your revised manuscript "A general prediction model for substrates of transport proteins" for publication as a Methods and Resources Article at PLOS Biology. Please accept my apologies for the delays that you have experienced during this round of the peer review process. This revised version of your manuscript has been evaluated by the PLOS Biology editors, the Academic Editor and two of the original reviewers.

Based on the reviews, I am pleased to say that we are likely to accept this manuscript for publication, provided you satisfactorily address the following editorial and data-related requests that I have provided below (A-E): 

(A) We would like to suggest the following modification to the title:

“SPOT: a machine learning model that predicts specific substrates for transport proteins” 

(B) You may be aware of the PLOS Data Policy, which requires that all data be made available without restriction: http://journals.plos.org/plosbiology/s/data-availability. For more information, please also see this editorial: http://dx.doi.org/10.1371/journal.pbio.1001797

-Supplementary files (e.g., excel). Please ensure that all data files are uploaded as 'Supporting Information' and are invariably referred to (in the manuscript, figure legends, and the Description field when uploading your files) using the following format verbatim: S1 Data, S2 Data, etc. Multiple panels of a single or even several figures can be included as multiple sheets in one excel file that is saved using exactly the following convention: S1_Data.xlsx (using an underscore).

-Deposition in a publicly available repository. Please also provide the accession code or a reviewer link so that we may view your data before publication. 

Regardless of the method selected, please ensure that you provide the individual numerical values that underlie the summary data displayed in the following figure panels as they are essential for readers to assess your analysis and to reproduce it. Please accept my apologies if this data is already included in your Zenodo deposition (https://doi.org/10.5281/zenodo.8358683):

Figure 2A-B, 3A-B, 4A-B, 5A-B, S1-6 

(C) Please also ensure that each of the relevant figure legends in your manuscript include information on *WHERE THE UNDERLYING DATA CAN BE FOUND*, and ensure your supplemental data file/s has a legend.

(D) Please ensure that your Data Statement in the submission system accurately describes where your data can be found and is in final format, as it will be published as written there. 

(E) Please note that we cannot accept sole deposition of your code in GitHub, as this could be changed after publication. However, you can archive this version of your publicly available GitHub code to Zenodo. Once you do this, it will generate a DOI number, which you will need to provide in the Data Accessibility Statement (you are welcome to also provide the GitHub access information). See the process for doing this here: https://docs.github.com/en/repositories/archiving-a-github-repository/referencing-and-citing-content

We expect to receive your revised manuscript within two weeks. 

*Published Peer Review History*

*Press*

Best wishes,

Richard

Richard Hodge, PhD

rhodge@plos.org

Reviewer remarks:

Reviewer #2: The authors have submitted a revision that is highly responsive to the previous concerns raised, not only by myself but also the other two reviewers. All issues have been addressed in a thoughtful and rigorous manner, and various additional analyses have been added that further strengthen this study. I have no further comments.

Reviewer #3: The authors have appropriately replied to my points, and have apparently fixed the problem with entered KEGG id's.

I ran the following 4 more tests for recently discovered transport functionalities:

https://elifesciences.org/articles/92615

reported in April 2024 that D-serine is a substrate of human SLC5A8 (SMCT1) and SLC7A10 (Asc-1).

(1) I tested Spot with SLC5A8 and C00740: this gave a prediction score of 0.050 which is incorrect.

(2) Then I tested Spot with SLC7A10 and C00740: this gave a prediction score of 0.68 which is correct.

PMID: 38507452 reported in March 2024 that the orphan lysosomal SLC, the Major Facilitator Superfamily Domain-containing Protein 1 (MFSD1)

transports lysine/arginine/histidine-containing dipeptides, e.g. the dipeptide KP

(3) I tested Spot with Mfsd1 and the Inchi-String of Lysyl-proline:InChI=1S/C11H21N3O3/c12-6-2-1-4-8(13)10(15)14-7-3-5-9(14)11(16)17/h8-9H,1-7,12-13H2,(H,16,17)/p+1

this gave a prediction score of 0.12 which is incorrect

https://www.biorxiv.org/content/10.1101/2024.03.22.586268v1.full.pdf

reports that VC0430 from Vibrio cholerae is a monomeric high affinity Transporter for L-glutamate.

(4) I used the sequence https://www.ncbi.nlm.nih.gov/protein/AAF93603.1

that appears to belong to VC0430 and the KEGG ID of l-Glutamate: C00025

I gave this to Spot. This gave a prediction score of 0.60 which is correct.

I believe that the authors of Spot overall did a very good job. The tool seems to work surprisingly well for compounds and transporters that are reasonably well represented by their training data set. My 4 new tests show that transferability to orphan transporters and new transport scenarios gave mixed results. Also, my previous tests only were successful for 4 out of 6 cases. Thus, users of Spot who want to discover truly novel substrates may experience that the reported 92% accuracy does not extend into such areas.

But I still think Spot is a good and useful addition to the literature and I am supportive of acceptance.

---

## [Editor Report · Decision Letter 2]

13 Aug 2024

Dear Dr Lercher,

On behalf of my colleagues and the Academic Editor, Raimund Dutzler, I am pleased to say that we can accept your manuscript for publication, provided you address any remaining formatting and reporting issues. These will be detailed in an email you should receive within 2-3 business days from our colleagues in the journal operations team; no action is required from you until then. Please note that we will not be able to formally accept your manuscript and schedule it for publication until you have completed any requested changes.

PRESS

Best wishes,

Richard 

Richard Hodge, PhD

rhodge@plos.org

PLOS
